# Mutational Dynamics Related to Antibiotic Resistance in *M. tuberculosis* Isolates from Serial Samples of Patients with Tuberculosis and Type 2 Diabetes Mellitus

**DOI:** 10.3390/microorganisms12020324

**Published:** 2024-02-03

**Authors:** Gustavo A. Bermúdez-Hernández, Damián Pérez-Martínez, Maria Cristina Ortiz-León, Raquel Muñiz-Salazar, Cuauhtemoc Licona-Cassani, Roberto Zenteno-Cuevas

**Affiliations:** 1Biomedical Sciences Doctoral Program, Institute of Health Sciences, University of Veracruz, Xalapa 91190, Veracruz, Mexico; dcsgabh@gmail.com; 2Institute of Public Health, University of Veracruz, Xalapa 91190, Veracruz, Mexico; damperez@uv.mx (D.P.-M.); cortiz@uv.mx (M.C.O.-L.); 3School of Health Sciences, Autonomous University of Baja California, Ensenada 22860, Baja California, Mexico; ramusal@uabc.edu.mx; 4Monterrey Institute of Technology, School of Engineering and Sciences, Monterrey 64700, Nuevo León, Mexico; clicona@tec.mx

**Keywords:** tuberculosis, drug resistance, diabetes mellitus, host, influence, polymorphisms

## Abstract

Genetic variation in tuberculosis is influenced by the host environment, patients with comorbidity, and tuberculosis–type 2 diabetes mellitus (TB-T2DM) and implies a higher risk of treatment failure and development of drug resistance. Considering the above, this study aimed to evaluate the influence of T2DM on the dynamic of polymorphisms related to antibiotic resistance in TB. Fifty individuals with TB-T2DM and TB were initially characterized, and serial isolates of 29 of these individuals were recovered on day 0 (diagnosis), 30, and 60. Genomes were sequenced, variants related to phylogeny and drug resistance analyzed, and mutation rates calculated and compared between groups. Lineage X was predominant. At day 0 (collection), almost all isolates from the TB group were sensitive, apart from four isolates from the TB-T2DM group showing the mutation *katG* S315T, from which one isolate had the mutations *rpoB* S450L, *gyrA* A90G, and *gyrA* D94G. This pattern was observed in a second isolate at day 30. The results provide a first overview of the dynamics of mutations in resistance genes from individuals with TB-T2DM, describing an early development of resistance to isoniazid and a rapid evolution of resistance to other drugs. Although preliminary, these results help to explain the increased risk of drug resistance in individuals with TB and T2DM.

## 1. Introduction

According to the World Health Organization (WHO), 10.6 million people developed tuberculosis (TB) in 2021, and the disease therefore represents a serious health problem [1]. Drug resistance (DR), HIV, and type 2 diabetes mellitus (T2DM) are factors that contribute to maintaining the condition of TB as a serious health concern. According to the Global Tuberculosis Report 2022, diabetes ranked fifth in the risk factors associated with incident cases of TB in 2021.

Tuberculosis is a curable disease, the treatment regimen of which is known as directly observed therapy (DOTS), in which the medical health worker supervises the ingestion of four antibiotics: isoniazid (H), rifampicin (R), pyrazinamide (Z), and ethambutol over four to six months. If suppression of infection is not observed, the resistance profile to first-line drugs is then determined and a new group of drugs is administered within the second-line regimen. Inadequate drug regimens, poor adherence to or abandonment of treatment, human immunodeficiency virus (HIV), and T2DM are all factors associated with the development of drug-resistant TB [1].

In this regard, T2DM increases 3.1-fold the risk of developing TB [2,3], and the TB-T2DM binomial has been recognized as a risk factor in unfavorable treatment outcomes, increasing the probability of failure, relapse, or death [4,5]. Individuals with this comorbidity have a 4.7-fold greater risk of becoming single-drug resistant, and a 2.8 to 3.5-fold increased risk of becoming multidrug resistant (TB-MDR) [6,7,8,9]. It also acts to promote the elongation of active infection and proliferation of TB bacilli [10,11]. This unfavorable evolution has been partially explained by the decreased plasma concentrations of anti-TB drugs [12,13,14], interference by drugs used for glycemic control [15], and immunological alterations [16,17,18,19,20,21]. Regrettably, growth in the incidence of type 2 diabetes mellitus in the coming years is mainly expected to affect low-income countries, which also have the highest levels of tuberculosis and drug-resistant tuberculosis, so the frequency of this binomial is expected to increase significantly in the coming years, hindering the possibility for controlling TB in the next decade.

There are three main factors related to the active infection of TB: susceptibility of the host, characteristics associated with *M. tuberculosis*, and the conditions of the host that modulate the progression of the infection. In this sense, many genetic markers such as the human leukocyte antigen (HLA) and non-HLA genes such as the killer immunoglobulin-like receptor (KIR), Toll-like receptors, cytokine/chemokines and their receptors, vitamin D receptor (VDR), and SLC11A1 have been related to the susceptibility of infection by tuberculosis [22]. Regarding the host condition, some markers have been associated with increased susceptibility to active TB in HIV-seropositive patients, such as the polymorphism Asp299Gly in the *tlr4* gene [23], and mutations at positions 333 and 637 of the *tap1* gene have been recognized as risk factors for developing TB coinfection in HIV-positive individuals [24].

In this regard, the influence of host conditions has also been described as a determinant in the development of polymorphisms in *M. tuberculosis*, promoting survival and drug resistance through the induction of specific mutations in specific genes. It is important to consider that single-nucleotide polymorphisms (SNPs) are the main source of variations in *M. tuberculosis* (*Mtb*) [25] and, thanks to whole-genome sequencing (WGS) analysis of serial sputum samples taken from patients affected with TB, it has been confirmed that genetic variations in *M. tuberculosis* are influenced by the host environment [26,27], and it has also been possible to identify the occurrence of specific polymorphisms related to virulence and resistance in individuals with specific characteristics. It has been reported that rifampicin-resistant *M. tuberculosis* strains isolated from HIV-coinfected patients carry more resistance-conferring mutations in *rpoB* and are associated with low fitness in the absence of the drug [28]. It has recently been described that some specific variations are associated with T2DM in DNA reparation genes in *M. tuberculosis* [29]; however, no differences have been described in the polymorphisms observed in canonical genes related to antibiotic resistance in individuals with the binomial TB-T2DM [30], suggesting that the high frequency of resistance observed in individuals with T2DM may be due to the rapid and early occurrence of variants in the resistance genes that allow rapid development [31]. The present study therefore aimed to evaluate the influence of T2DM in the dynamics of polymorphism generation related to the development of antibiotic resistance in *M. tuberculosis*.

## 2. Methods

### 2.1. Population

This is a descriptive longitudinal study, which included genomes of *M. tuberculosis* complex (MTBC) taken from a random selection of patients diagnosed with pulmonary TB from the regions of Xalapa and Cordoba and clinically confirmed by the Veracruz Health Secretary from the period 2020–2022. A subset of individuals with T2DM-TB and TB were further selected and followed up from the start until the 60th day of treatment. Sputum samples were collected at different times: T0 at the time of diagnosis and initiation of treatment and T1 and T2 at 30 and 60 days after diagnosis, respectively.

### 2.2. DNA Extraction and Whole Genome Sequencing

MTB strains were isolated in LJ media and genomic DNA was extracted and purified following the CTAB method [32]. The DNA was quantified using a nanodrop (Thermo Scientific, Waltham, MA, USA), with subsequent adjustment to a concentration of 0.2 ng/µL. The WGS libraries were prepared according to the Nextera XT (Illumina, San Diego, CA, USA) protocol, using 1 ng/mL of DNA previously quantified by a Qubit fluorometer (Invitrogen, Waltham, CA, USA). Quality control of the libraries was conducted using TapeStation (Agilent Genomics, Santa Clara, CA, USA), which were normalized and sequenced using NexSeq 500 (Illumina, San Diego, CA, USA) in a 2 × 150 paired-end format. Genome sequences are available under the bioproject number PRJNA1041872.

### 2.3. Bioinformatics Analysis and Drug Resistance Prediction

The WGS reads were processed with Fastqc and Trimmomatic (V.32). Kraken2 (v2.1.2) was used together with Seqtk v1.3 to clean the genomes of non-MTb reads, and reads that belonged to the MTBC species were subsequently analyzed with the pipeline MTBseq [33]. Mapping and variant calling (SNPs and INDELS) were performed with SAMtools [34]. Variants in 20 reads and at ≥90% frequency were called fixed SNPs.

Variants in at least 10 reads with a frequency of ≥10% to ≤90% were called no-fixed SNPs and were used to detect antibiotic resistance. The canonical resistance genes studied were *rpoB* and *rpoC* for rifampicin (R), *katG, ahpC,* and *inhA* for isoniazid (H), *pncA,* and *rpsA* for pyrazinamide (K), and *embC*, *embA,* and *embB* for ethambutol (E). With respect to the second-line drugs, the genes included were *gyrA* and *gyrB* for fluoroquinolones and *rrs, eis,* and *tlyA* for kanamycin, capreomycin, amikacin, and streptomycin. Each SNP was further compared to the Phyresse catalog and other reports [35,36]. The resulting information was used to create a database that included detailed information regarding the SNPs and INDELs, and the metadata associated with each patient.

### 2.4. Phylogenetic Analysis and Cluster Identification

Phylogeny was created with a concatenated alignment made with fixed SNPs of all clinical isolates recovered. Fixed SNPs (those detected at a minimum frequency of >90%) were used to classify samples into lineages and to obtain the final alignment for the detection of clonal complexes, while variable SNPs (those detected at a frequency of 10–90%) were used to detect resistances. To avoid a false positive variant calling, SNPs annotated in PE/PPE/PGRS and phage genes were removed. In addition, SNPs (both fixed and variable) detected within insertion sequences, INDELS, and high-density regions (>3 SNPs in 10 bp) were also discarded from this analysis. This alignment was used to infer phylogeny using RAxML v8.2.4, considering the maximum likelihood phylogenetic approach, applying a general time-reversible model of nucleotide substitution with a gamma distribution (GTR + GAMMA). The tree was finally visualized in iTOL V. 4. [37].

Strains were classified according to the presence of 62 phylogenetic variants associated with lineages and sublineages, as proposed by Coll et al. [38].

### 2.5. Allelic Frequency, Mutation Rate, and Statistical Analysis

The SNPs identified in resistance genes were compared within and between groups. Their allelic frequency was determined and the mutation rate of the genomes of patients with at least two isolates was calculated using the following formula:μ=m/[N xt/g)]

The mutation rate (*μ*) was determined by dividing the number of observed SNPs (*m*) by the genome size (*N*) multiplied by the number of generations (t/g). *N* was determined based on 99% coverage of a 4.4 Mb genome (4.4 × 10^6^), *t* is the duration of each infection in hours, and *g* is the generation time in hours (20 h per cycle) [39].

Chi-squared and Mann–Whitney U tests were performed to identify differences between groups. Comparison of means was conducted using the Student’s *t*-test. A *p*-value < 0.05 was considered significant. The software IBM SPSS V. 25 was used for statistical processing [40].

## 3. Results

### 3.1. Population Characteristics and Genotypic Resistance

The initial group of the study included fifty *M. tuberculosis* isolates recovered from the same number of individuals; all patients were diagnosed as TB cases, according to clinical exams and positive acid-fast bacilloscopy. Distribution according to sex was 37 females and 13 males, the mean age was 45 years (±4.4), and 26 (52%) isolates presented T2DM-TB.

Genotypic resistance was observed in eleven patients (42%) with T2DM-TB and eight (33%) with TB. A specific description of genotypic resistance prediction against first- and second-line drugs is presented in Table 1. It was not possible to determine the phenotypic profile of resistance because the patient must show clinical signs of resistance, according to the Mexican national standard, to perform the tests to confirm this condition. It must also be considered that all individuals have a recent TB diagnostic; therefore, no phenotypic drug-resistant data were available from the patients.

### 3.2. Characterization of Variants Associated with Resistance and Phylogeny

A total of 36 SNPs were detected among the genomes sequenced; 22 were mainly observed in individuals with T2DM-TB and 14 in individuals with TB.

Five mutations related to drug resistance genes were observed. The most common mutations related to resistance to isoniazid were *katG* S315T, observed in nine isolates from individuals with DM2-TB and two patients with TB, followed by *fabG1* C15T, detected in one individual with T2DM-TB and three individuals with TB. The most frequent mutation associated with resistance to rifampicin was *rpoB* S450L, found in four isolates from individuals with DM2-TB and in one isolate from one individual with TB. Mutations related to resistance against pyrazinamide were observed less frequently; mutation *pncA* L120P was observed in two isolates from individuals with T2DM-TB and only one isolate from one individual with TB. With respect to resistance against fluoroquinolone, variation *gyrA D94G* was observed in two isolates from the same number of individuals with T2DM-TB and one isolate from one individual with TB. A second mutation in this gene, *gyrA* A90V, was observed in one isolate from one individual with T2DM-TB.

Regarding the lineages observed in the isolates recovered, two main lineages (L) were identified: L4 was observed in forty-seven (94%) isolates and L1 was found in three isolates (6%). Within the L4 lineage, five sublineages were identified, the most abundant of which was X, observed in twenty-four (48%) isolates, followed by LAM in ten (20%), Haarlem in eight (16%), and Euro-American and H37Rv with three (6%) each. In the second lineage, L1, the EIA-Manila sublineage was observed in three isolates (Figure 1).

According to the characteristics of the individuals, the X-type sublineage was observed in sixteen (32%) isolates from the T2DM-TB group of individuals and eight (16%) isolates from the TB group of patients. Lineage LAM was observed in four (8%) patients from the T2DM-TB group and six (12%) from the TB group. H37Rv was found in one (2%) isolate from one individual with T2DM-TB and in two people with TB (4%), while Haarlem was observed in two patients with T2DM-TB (4%) and six (12%) individuals with TB. Finally, the Euro-American sublineage was found in one patient with TB (2%), and the EAI-Manila sublineage was observed in three individuals all with TB only.

### 3.3. Characterization of Serial Isolates: Mutations in Canonical Resistance Genes and Phylogeny

Adequate follow-up was achieved in fifteen individuals, of whom nine were people with T2DM-TB and six were individuals with TB. Considering the number of samples per type of individual, five individuals with T2DM-TB provided three serial samples (T0: Diagnosis, T1: 30th day, and T2: 60th day), while four individuals provided only two serial samples (T0 and T1). Of the six individuals with TB, two serial samples were recovered: four individuals contributed samples at T0 and T1 and two contributed samples at T0 and T2, considering that these patients did not provide a sample of sufficient quality to obtain the respective isolate at the 30-day time of visiting the medical service.

Genotypic resistance against drugs was observed only in four isolates, all belonging to the T2DM-TB group of individuals, and the evolution of mutations associated with the resistances is presented in Table 2. At the time of diagnosis (T0), the three isolates, 2021-011, 2021-025, and 2021-043, presented the *katG* S315T mutation and were therefore considered mono-resistant to isoniazid. In addition, one isolate, 2021-031, presented the mutation *katG* S315T, in combination with mutations *rpoB* S450L, *gyrA* A90G, and *gyrA* D94G, conferring the category of pre-XDR to this isolate. At T1 (30th day), samples 2021-011 and 2021-043 maintained the mutation *katG* S315T and the mono-resistance against isoniazid. The isolate 2021-031 maintained the same pattern of mutations associated with the pre-XDR condition, and interestingly, the isolate 2021-025 now shared the same polymorphisms observed in the 2021-031 isolate, conferring the pre-XDR condition. At T2 (60th day), all isolates kept the same mutations. Unfortunately, the conditions of the sputum sample recovered from the patient 2021-025 did not allow the acquisition of clinical isolation. Finally, is important to mention that the remaining isolates from individuals of both groups, T2DM-TB and TB, showed no changes and were therefore considered sensitive. 

The phylogenetic analysis of the serial samples showed a predominance of L4: the Euro-American sublineage (4.1) was identified in one isolate from a patient with T2DM-TB. The type-X sublineage was identified in ten individuals, and divided into two subgroups; 4.1.1, including two individuals with T2DM-TB and four with TB, and 4.1.1.3, observed in four isolates from individuals with T2DM-TB. Finally, the LAM sublineage was observed in four patients and divided into two subgroups: 4.3.4.2, included in only one isolate from the T2DM-TB group, and 4.3.3, found in three isolates from individuals with TB (Figure 1).

### 3.4. Total Number of Polymorphisms and Quantification of the Mutation Rate of Serial Isolates

The analysis of the total number of nonsynonymous SNPs in the genomes of the serial isolates showed a mean of 963 (SD = 50). According to the groups, it was 960 (SD = 48) in those individuals with TB-T2DM and 970 (SD = 50) with TB. According to the evolution of the infection, at T0, the mean number of SNPs was 960 (SD = 51); in the genomes of individuals with the binomial, it was 959.3 (SD = 44), and with TB it was 965 (SD = 69). At T1, the mean number was 970 (SD = 52), decreasing to 963 (SD = 45) in the genomes of individuals with the binomial and increasing to 979 (SD = 58) in individuals with TB only. For T2, a mean value of 958 (SD = 50) was observed, finding 959 (SD = 56) in individuals with the binomial and 955 (SD = 10) in individuals with TB only. No significant differences were observed between groups.

Using the T0 sample as a starting reference, 36 generations were counted for the *M. tuberculosis* isolates recovered at T1, while 72 generations were determined for T2. In this sense, the mutation rate for isolates recovered at T1 was 7.92 × 10^−3^ (SD = 4.22 × 10^−4^) while, for those genomes from patients with T2DM-TB, it was 7.86 × 10^−3^ (SD = 3.67 × 10^−4^), and from those with TB only, it was 7.99 × 10^−3^ (SD = 4.71 × 10^−4^). On the other hand, in the T2 isolates, the mean mutation rate was 1.56 × 10^−2^ (SD = 4.22 × 10^−4^), while in the TB genomes of individuals with T2DM-TB, it was 1.57 × 10^−2^ (SD = 9.2 × 10^−4^), and in patients with TB it was 1.56 × 10^−2^ (SD = 1.63 × 10^−4^).

### 3.5. Phylogenetic Relationship and Mutation Rates of Serial Isolates

The mutation rates observed in our study provide valuable insights into the genetic diversity of *M. tuberculosis* strains in individuals with T2DM-TB and with TB alone. The Euro-American sublineage displayed an overall mean mutation rate of 1.26 × 10^−2^, indicating a moderate level of genetic variation. Sublineages X (4.1.1) and X (4.1.13) exhibited slightly higher mutation rates (1.37 × 10^−2^ and 1.11 × 10^−2^, respectively), suggesting a greater propensity for genetic change. In contrast, sublineage LAM (4.3.3) showed a relatively lower mutation rate (1.08 × 10^−2^), indicating a more conserved genetic profile. Interestingly, only two sublineages, X (4.1.1) and LAM (4.3.3), were shared between individuals with T2DM-TB and TB, albeit with varying mutation rates. Table 3 shows the mutation rates for each group.

## 4. Discussion

After analysis of the canonical resistance genes in 50 individuals, we identified the most frequent mutations as *katG* S315T, conferring resistance to isoniazid, and *rpoB* S450L to rifampicin. This is consistent with previous reports [41,42,43] and could be due to the early appearance of these mutations in the genome of resistant strains [44], as well as the low fitness of these variants, which makes them more transmissible in bacteria [45]. We also observed a higher frequency of these SNPs in isolates from individuals with the T2DM-TB binomial, possibly because of the decreased plasma concentration of anti-TB drugs [12,13,14] and/or interference by drugs used for glycemic control [15].

Based on the findings of the phylogenetic analysis, sublineage X appears to be prevalent in the isolates analyzed. It was detected in 50% of the isolates recovered and in 90% of the isolates from the T2DM-TB group. This sublineage has been increasing significantly in Mexico in recent years and is frequently associated with drug resistance [43,46,47,48]. Indeed, more than 50% of the isolates within this sublineage demonstrated resistance, and the SNP *katG* S315T was the primary cause of resistance against isoniazid. More in-depth research is required to better understand the prevalence and mechanisms of dispersion of this sublineage among the Mexican population [49,50].

The analysis of the genome of serial isolates indicated that only individuals with TB-T2DM develop resistance and that isoniazid resistance at an early stage of the infection was mainly due to the presence of the *katG* S315T mutation. Two of the four TB-T2DM isolates were part of a cluster, suggesting a possible primary transmission of resistance. However, two isolates resistant to isoniazid were not part of this cluster, and one of these developed a set of mutations related to pre-extreme drug resistance within just 30 days. This finding highlights the accelerated selection process of mutations in resistance genes in individuals with TB-T2DM. It is therefore crucial to have a genomic surveillance system for individuals with T2DM, including early detection of TB and identification of mutations related to resistance against drugs.

The mutation rate of *M. tuberculosis* is minimized in populations that are well adapted to their environment [51]. However, it is recognized that conditions such as antimicrobial treatment and high glucose concentration in T2DM individuals can lead to the rapid induction, selection, and fixation of mutations in specific genes. By determining the mutation rate of the serial isolates, we obtain an overview of the mutation dynamics in some genes related to resistance. Although average mutation rates were similar among all groups, the most significant differences between groups were in the variants in those genes related to drug resistance in individuals with T2DM. Moreover, one isolate from the TB-T2DM individuals group showed a significant increase in mutation rates from 30 to 60 days after initiation of treatment. In this regard, chronic hyperglycemia due to poorly controlled diabetes may promote the accumulation of advanced glycation products (AGEs) and receptors for AGEs (RAGEs), mainly expressed in the lungs, the primary site of tuberculosis infection [52]. Activation of RAGEs leads to increased inflammation due to the production of reactive oxygen species and proinflammatory cytokines, such as IL-1β and IL17, resulting in defective phagocytosis [53], coupled with an increase in oxidative stress and changes in the distribution of anti-tuberculosis drugs [51]. These are factors that could modify the mutation rates of the bacterium [53], promote the appearance of SNPs associated with resistance, and partially explain why patients with T2DM present an increased risk of developing drug- and multidrug-resistant tuberculosis (MDR-TB) [6,7,8,9], as well as an elongated active period of TB [10,11].

This study was subject to some limitations due to the COVID-19 pandemic. Consequently, there was a limited number of isolates, and some patients were lost and could not be followed up. Despite these limitations, we recovered enough serial isolates to identify differences in the behavior of SNPs associated with resistance and to analyze mutation rates. This study focused on the impact of diabetes on the development of tuberculosis drug resistance, which had not previously been studied using WGS. The results, although preliminary, provide valuable insights into mutation rates and the behavior of resistance-associated SNPs.

## 5. Conclusions

In conclusion, no significant differences were observed between the TB-DM2 and TB groups in terms of the distribution of resistant isolates, and the high frequency of *katG S315T* and *rpoB S450L* SNPs highlights the importance of strengthening TB control and surveillance mechanisms in the context of T2DM. Analysis of the serial samples showed the early presence of polymorphisms associated with isoniazid resistance, which rapidly evolved to pre-XDR forms. The study provides an initial idea about the effect of the conditions of individuals with T2DM, which favors the early expression and fixation of variants associated with resistance that could explain the elevated risk of drug resistance described in individuals carrying the binomial. Confirmation of this influence could contribute to the establishment of new therapeutic, attention, and control models for tuberculosis in the context of T2DM.

## Figures and Tables

**Figure 1 microorganisms-12-00324-f001:**
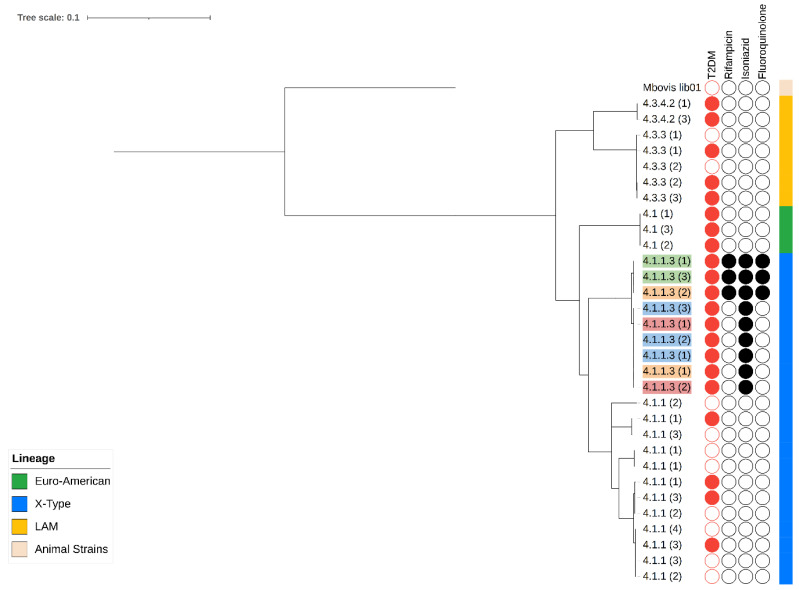
Phylogenetic tree and clinical characteristics of *M. tuberculosis* genomes from patients subjected to monitoring. Red circle: occurrence of T2DM. Black circle: genotypic resistance against rifampicin, isoniazid, and fluoroquinolones.

**Table 1 microorganisms-12-00324-t001:** Genotypic drug resistance prediction of *M. tuberculosis* isolates in serial samples.

Drug/Group of Isolates	TB 8 (%)	T2DM-TB 11 (%)
Isoniazid	5 (63)	10 (91)
Rifampicin	1 (13)	4 (26)
Pyrazinamide	1 (13)	2 (18)
Streptomycin	3 (38)	1 (9)
Ethionamide	3 (38)	1 (9)
Aminoglycosides	2 (25)	1 (9)
Fluoroquinolone	1 (13)	2 (18)

**Table 2 microorganisms-12-00324-t002:** Distribution of SNPs associated with resistance in serial T2DM-TB isolates.

Patient	T0 (0 Day)	T1 (30th Day)	T2 (60th Day)
2021-011	*katG* S315T	*katG* S315T	*katG* S315T
2021-025	*katG* S315T	*katG* S315T	--
		*rpoB* S450L	
		*gyrA* A90G	
		*gyrA* D94G	
2021-031	*katG* S315T	*katG* S315T	*katG* S315T
	*rpoB* S450L	*rpoB* S450L	*rpoB* S450L
	*gyrA* A90G	*gyrA* A90G	*gyrA* A90G
	*gyrA* D94G	*gyrA* D94G	*gyrA* D94G
2021-043	*katG* S315T	*katG* S315T	*katG* S315T

**Table 3 microorganisms-12-00324-t003:** Relationships between individuals with DM2-TB and TB in terms of lineage and mutation rate.

Sublineage	Classification	Mean Mutation Rate	Mutation Rate in DM2-TB	Mutation Rate in TB
Euro-Americano	4.1	1.26 × 10^−2^	1.26 × 10^−2^	0
X-type	4.1.1	1.37 × 10^−2^	1.61 × 10^−2^	1.33 × 10^−2^
X-type	4.1.1.3	1.11 × 10^−2^	1.11 × 10^−2^	0
LAM	4.3.3	1.08 × 10^−2^	1.20 × 10^−2^	7.23 × 10^−3^
LAM	4.3.4.2	1.39 × 10^−2^	1.39 × 10^−2^	0

## Data Availability

Genome sequences are available under the bioproject number PRJNA1041872.

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
