# Peer review of "Mutational Dynamics Related to Antibiotic Resistance in *M. tuberculosis* Isolates from Serial Samples of Patients with Tuberculosis and Type 2 Diabetes Mellitus"

_microorganisms, 2024, doi:10.3390/microorganisms12020324_

Round 1

Reviewer 1 Report

Comments and Suggestions for Authors

1. Since the author is studying drug resistance, it is recommended to add treatment instructions for the disease in the introduction , including treatment duration, type of medication, etc

2. Chi square, Mann Whitney U tests and Student's t-test were performed, but I'm sorry I couldn't find these statistical results

3. The conclusion section shows that this study does not seem to have significant clinical drug guidance significance

4. Suggestion: In addition to classic papers, update the cited references to those published in recent years

Comments on the Quality of English Language

In terms of writing proficiency, there is no ambiguity when reading, but after revision, it can be more professional, concise, and intuitive.

Author Response

Thanks to the reviewer for the comments and suggestions.

  1. Since the author is studying drug resistance, it is recommended to add treatment instructions for the disease in the introduction , including treatment duration, type of medication, etc

A short description of the treatment, considering duration, drugs and factors related with drug resistance was included in the introduction.

  1. Chi square, Mann Whitney U tests and Student's t-test were performed, but I'm sorry I couldn't find these statistical results

Despite of the interesting results obtained, unfortunately, no statistical significances were observed.

  1. The conclusion section shows that this study does not seem to have significant clinical drug guidance significance

Regrettably, the results obtained in this work were preliminary and not allow to state a clinical guidance o final recommendation, this was now included as a “future confirmation” in the last lines of the conclusion section.L438-441.

  1. Suggestion: In addition to classic papers, update the cited references to those published in A recent years

An additional set of actual publications related with the influence of diabetes on tuberculosis, were now included in the introduction section.

Reviewer 2 Report

Comments and Suggestions for Authors

On request of Microorganisms, I have revised the manuscript titled “Mutational Dynamics Related to Antibiotic Resistance in M. tuberculosis Isolates from Serial Samples of Patients with Tuberculosis-Type 2 Diabetes Mellitus” by Bermúdez Hernández Gustavo Adolfo and co-workers.

The main scope of this work was to evaluate the influence of T2DM on the dynamic of polymorphisms related to antibiotic resistance in TB. To this end, the authors considered 50 individuals with TB-T2DM and TB, which were initially characterized, and from which serial isolates of 29 individuals were recovered at day 0 (diagnosis), 30 and 60.

COMMENTS

Although preliminary, the findings reported here could help to explain the increased risk of drug-resistance in individuals with TB and T2DM.

Some minor issues should be addressed before publication.

In the abstract (line 21), I suggest using the past tense in place of present.

Line 43. Please, add the reference.

Lines 115 and 117. Please, insert a space between the numbers and ng. Check all manuscript and solve similar problems.

Introduction and discussion should be improved.

A conclusion section should be inserted in the manuscript.

Author Response

Thanks to the reviewer for the comments and suggestions.

1. In the abstract (line 21), I suggest using the past tense in place of present.

Thanks for the observation, this was attended.

2. Line 43. Please, add the reference.

Reference was included.

3. Lines 115 and 117. Please, insert a space between the numbers and ng. Check all manuscript and solve similar problems.

This error was corrected, and manuscript revised.

4. Introduction and discussion should be improved.

A new paragraph was included in the introduction and a revision of the discussion was done and several adjustments made.

5. A conclusion section should be inserted in the manuscript.

The conclusion section was remarked in this new version.